# In the context of romantic attraction, beautification can increase assertiveness in women

Khandis R. Blake[1,2]*, Robert Brooks[1], Lindsie C. Arthur[3], Thomas F. Denson[4]

**1** Evolution and Ecology Research Centre, School of Biological, Environmental and Earth Sciences, The University of New South Wales, Sydney, New South Wales, Australia, **2** Melbourne School of Psychological Sciences, The University of Melbourne, Melbourne, Victoria, Australia, **3** Department of Psychological and Behavioural Science, London School of Economics, London, United Kingdom, **4** School of Psychology, The University of New South Wales, Sydney, New South Wales, Australia

\* k.blake@unsw.edu.au

**Data Availability Statement:** All data and materials and correlations between variables are available on the Open Science Framework (OSF; https://osf.io/n7tyc/) and all measures and manipulations are also disclosed in the manuscript.

## Abstract

Can beautification empower women to act assertively? Some women report that beautification is an agentic and assertive act, whereas others find beautification to be oppressive and disempowering. To disentangle these effects, in the context of romantic attraction we conducted the first experimental tests of beautification—on psychological and behavioral assertiveness. Experiment 1 ($N$ = 145) utilized a between-subjects design in which women used their own clothing, make-up, and accessories to adjust their appearance as they normally would for a "hot date" (beautification condition) or a casual day at home with friends (control condition). We measured implicit, explicit, and behavioral assertiveness, as well as positive affect and sexual motivation. Experiment 2 ($N$ = 40) sought to conceptually replicate Experiment 1 using a within-subject design and different measures of assertiveness. Women completed measures of explicit assertiveness and assertive behavioral intentions in three domains, in whatever clothing they were wearing that day then again after extensively beautifying their appearance. In Experiment 1, we found that women demonstrated higher psychological assertiveness after beautifying their appearance, and that high sexual motivation mediated the effect of beautification on assertive behavior. All effects were independent of positive affect. Experiment 2 partially replicated Experiment 1. These experiments provide novel insight into the effects of women's appearance-enhancing behaviors on assertiveness by providing evidence that beautification may positively affect assertiveness in women under some circumstances.

## Introduction

Women's preoccupation with their physical appearance and their consumption of beauty products presents a paradox. On the one hand, scholars highlight that women are pressured to beautify their appearance from a very young age [1]. They also experience pressure to consider their physical attributes as representing their greatest value as a person [2]. These pressures are

**Funding:** The current work was funded by a Future Fellowship (Australian Research Council) to TFD and a Discovery Project (Australian Research Council) to RB (https://www.arc.gov.au/). The funders had no role in study design, data collection and analysis, decision to publish, or preparation of the manuscript.

**Competing interests:** The authors have declared that no competing interests exist.

associated with numerous poor mental health outcomes [3], indicating that women's preoccupation with their appearance can be both harmful and potentially disempowering. On the other hand, some women claim that beautifying their appearance, especially to enhance their attraction to potential romantic partners, makes them feel empowered, strong, and independent [4]. Whether these claims of empowerment are authentic, or are instead false reports, raises the question: Is beautification—the act of enhancing one's physical appearance to look more beautiful—empowering or oppressive for women? In the current research, we attempt to shed light on this question by investigating the conditions under which beautification increases assertiveness in women.

## The beautification paradox: Is beautification oppressing or empowering?

Two divergent perspectives attempt to explain the effects of beautification on female psychology. The dominant perspective derives from a sizeable body of work on objectification theory [2], which holds that beautification is oppressive and disempowering to women. Objectification theory posits that power imbalances between men and women create a cultural environment where women are constantly subject to a sexualized male gaze. In other words, women are constantly looked upon and evaluated for their sexual and physical attractiveness to men (sexually objectification; [2]). As a result of sexual objectification, women internalize a third-person perspective of themselves as objects, whose sole value derives from their physical characteristics (i.e., they self-objectify; [2]).

According to the framework of objectification theory, beautification can be one of many ways that women self-objectify. Self-objectification has numerous negative consequences for women, including habitual body monitoring, body shame and appearance anxiety, reduced peak motivational states, and diminished interoceptive bodily awareness [2,3]. These consequences have negative effects for women's mental health, accounting for their higher rates of unipolar depression, sexual dysfunction, and eating disorders, compared to men [2,5,6]. Because of these negative psychological and behavioral effects, objectification theory reasons that beautification is part of a broader phenomenon that ultimately harms women. The implication is that when women engage in beautification, they are not only risking psychological harm, but are internalizing cultural conditions where women hold lower status compared with men [7].

An alternative perspective considers whether beautification might empower women in some circumstances, and derives from sociometer theory [8]. Sociometer theory argues that an individual's self-esteem depends upon the degree to which they are valued as a relational partner. People obtain high self-esteem when they perceive that they have been accepted by others in important domains. One domain where self-esteem is especially likely to be derived is the domain of physical attractiveness. Given the premium on female attractiveness across cultures [9], especially in the context of romantic relationships, this domain is especially relevant for women.

Studies indicate that nearly 25% of the variability in women's global self-esteem derives from their physical appearance [10]. Thus, beautification may be empowering to women inasmuch as it raises women's self-esteem in important domains. Consistent with this notion, recent work indicates that being sexually objectified by a romantic partner can have positive implications for women's self-esteem over time [11], and that self-objectification can raise self- efficacy and wellbeing in domains unrelated to appearance [12]. This work implies is that there may be tangible psychological benefits to beautification that objectification theory has overlooked.

## Beautification effects on women's agency

How might we investigate whether beautification is empowering or oppressive? At the heart of this beautification paradox and second- and third-wave feminist disagreements on female beauty practices is whether women's appearance-enhancement can be an authentic expression of female agency. Agency is a topic of investigation in a variety of psychological fields and refers to the capacity to exert control over the nature and quality of one's life [13]. Agentic characteristics allow individualistic action aimed at self-protection, self-expansion, and self-assertion. They are expressed in achievement-oriented behavior associated with empowerment, such as competence and intentionality [14,15].

Focusing on whether beautification increases or decreases agency may provide needed insight into the beautification paradox. Agentic characteristics are directly related to feelings of empowerment and assertiveness, and people who are empowered behave more assertively and with higher agency [15]. Thus, if beautification empowers women as sociometer theory predicts, then beautification should be associated with characteristics reflective of high agency, such as assertiveness. However, if beautification disempowers and oppresses women as predicted by objectification theory, then beautification should be associated with characteristics of low agency, such as low assertiveness.

Very little previous research has examined the relationship between beautification and agency. Insights into their covariation; however, can be derived by examining the relationship between agency and phenomena related to beautification, such as self-objectification and self-sexualization (favoring sexual self-objectification; [16]). This body of work suggests that there is some support for the notion that beautification and agency positively covary. Yet, to confuse matters, there is also support for the notion that the covariance between the two phenomena is negative.

Evidence suggestive of a positive covariation between beautification and agency derives from some feminist scholarship, which argues that self-sexualization elevates agency by allowing women to transgress stereotypically chaste expressions of female sexuality [17]. Consistent with this notion, some women report that self-sexualizing makes them feel assertive and self-efficacious, as well as strong, independent, and powerful [4,18–20]. Enhancing one's attractiveness through cosmetics also offers considerable social and economic benefits to women, including higher tips at work and perceptions of increased intelligence [21,22]. Likewise, tentative evidence suggests that attractiveness and assertiveness can co-occur [22], and that self-objectification can increase self-esteem in particular domains [12]. Taken together, these data suggest that appearance-enhancement, either through beautification, self-sexualization, or self-objectification, could elevate agentic characteristics and confer some benefits.

Yet not all prior research supports the notion that beautification might increase agentic characteristics. Although beautification can be associated with high agency, and self-objectification with domain-contingent self-esteem, self-objectification is also frequently shown to *reduce* women's ability for agentic action [23]. Likewise, self-objectification is positively correlated with reduced cognitive performance under some circumstances [24], and perhaps more worryingly, with gender system justification [7]. Enjoying self-sexualization and using cosmetics are also both positively correlated with ambivalent sexism in some cultures [25,26]. These findings suggest that beautification ultimately disempowers women and exemplifies system-justifying behavior. They also raise the possibility that beautification-induced feelings of agency are false reports.

If beautification is in fact disempowering to women, as objectification theory suggests, why might some women report that beautification is empowering? The most probable explanation for these ostensibly false reports is that they may reflect elevated positive mood, rather than

authentically high agency, per se. Both beautification and sexualization increase women's ratings of their own attractiveness and ratings of how attractive they are to others [27,28], and it is well known that people experience considerable benefits from fulfilling cultural attractiveness ideals [9]. If women are reporting higher assertiveness and self-efficacy from beautification, but truly what they are experiencing is merely positive mood, it is important to ensure that measures of agency are disambiguated from mood. If a positive relationship between beautification and assertiveness is not robust to statistically controlling for positive mood, then this finding would support the notion that beautification does not increase agency. Alternatively, if the relationship between beautification and assertiveness is robust to mood, this would provide more convincing support of a positive relationship between the two phenomena.

## The current experiments

Evidence associating beautification with female agency is mixed. We aimed to shed light on this phenomenon by experimentally disentangling the relationship between beautification and assertiveness. Because attractiveness in the domain of romantic relationships is a domain where women are especially likely to derive self-esteem [8,11], we focused on beautification in the context of a potential romantic relationship. Using between- and within-subjects designs, we tested the extent to which experimentally manipulated beautification increases assertiveness—a manifestation of agency—in women. To ensure our experimental tests were rigorous, we used explicit, implicit, and behavioral indicators of assertiveness. Given that some scholars posit that women may report empowerment from beautification, but actually may be experiencing higher positive mood and not actual empowerment [4], we also measured the effect of beautification on assertiveness when controlling for positive mood.

We also examined the relationship between beautification and assertiveness, controlling for sexual motivation. Theorists argue that some forms of beautification elevate agency because they are sexually empowering [29]. This idea, which is consistent with state assertiveness positively correlating with state sexual motivation in women [30], raises the possibility that beautification will only confer elevated agency to the extent that it increases women's sexual motivation. To see if the effect of beautification on assertiveness was due to increased sexual motivation, in Experiment 1 we examined the relationship between these variables in an exploratory fashion. In Experiment 2, we sought to reproduce and extend the findings from Experiment 1. Finally, to control for relevant individual differences, in Experiment 2 we examined and controlled for trait self-objectification. Scholars have noted a considerable conceptual murkiness between self-objectification and related phenomena [16], and we aimed to control for trait differences in the tendency to self-objectify to provide a crisper dissection of resulting effects.

## Experiment 1

Experiment 1 utilized explicit, implicit, and behavioral measures of assertiveness to test whether beautification—in the context of romantic attraction—elevated assertiveness in women. Participants altered their appearance to suit a "hot date" (beautification condition) or an afternoon at home with friends (control condition). We then examined assertiveness using explicit and implicit measures, and for a subgroup of the sample, we assessed their behavioral assertiveness in a mock job interview. Informed by sociometer theory [8] and objectification theory [2], we tested whether beautification would elevate assertiveness, sexual motivation, and positive affect, and whether the effect of beautification on assertiveness would be independent of positive affect.

## Method

### Participants and design

One hundred and fifty-eight women ($M_{Age}$ = 19.58, $SD$ = 3.40) were recruited for a study on fashion and psychology from University of New South Wales Sydney as part of course require-ments or in exchange for AUD$15. Pre-screening requirements were English language profi-ciency and 18–40 years of age. The University of New South Wales Sydney Human Research Advisory Ethics Panel approved the protocol. One participant was excluded due to extensive missing data, two for refusing to comply with study instructions, eight for failing the attention check, and two for receiving both sets of manipulation instructions, leaving $N$ = 145 ($M_{Age}$ = 19.58, $SD$ = 3.35). There were no other exclusions. Sample size was determined via an a priori power analysis in GPower 3.1.9.2 which showed that a sample of 142 participants achieved power at 0.80 to detect a medium effect for condition ($d$ = 0.50, α = .05, two-tailed) and recruitment stopped when the required sample size, plus 10% leeway for withdrawals, was reached. Participants were randomly assigned to the control or beautification condition in a between-subjects design ($n_{Control}$ = 78, $n_{Beautification}$ = 67). The online pre-screening survey randomly allocated 113 women into the control condition and 106 women into the sexualized clothing condition. Differences in participation between conditions were non-significant, χ2 (1, N = 219) = 0.83, p = .363.

Thirty-four percent (33.8%) of participants were White, 5.5% were mixed White/Asian, 9% were Southern Asian, 40% were Eastern Asian, 3.4% were Middle Eastern, 7.6% were South-East Asian, and 0.7% were Polynesian. The majority of participants (81.4%) were Australian citizens. All participants provided written informed consent and were debriefed in accordance with the Declaration of Helsinki. All data and materials and correlations between variables are available on the Open Science Framework (OSF; https://osf.io/n7tyc/) and all measures and manipulations are also disclosed here. In addition to the measures listed below, we also mea-sured estradiol both at baseline and at the completion of the experiment. We hoped to establish a biopsychosocial pathway to female assertiveness. Estradiol neither mediated nor moderated the effect of beautification on any assertiveness outcome (all mediation confidence intervals contained zero and all moderation $p$-values were > .10). Interested readers should consult the OSF (https://osf.io/n7tyc/) for further details.

### Procedure and materials

After completing an online prescreening survey, participants were instructed to bring at least one additional clothing outfit to their laboratory session, depending on condition. Upon arrival to their session, participants were verbally briefed then escorted to a change-room where they were given 10–15 minutes to comply with the condition instructions. After this time elapsed, participants returned to the testing room and completed the explicit assertiveness measure, an affect scale, and the implicit assertiveness measure. They were then invited to par-ticipate in an additional task for extra credit or AUD$5 where they auditioned as a candidate for a hypothetical job in a video-recorded interview (the behavioral assertiveness measure).

**Beautification manipulation.** To manipulate beautification in the context of romantic attraction, participants were instructed to bring a full change of clothing (including shoes, jewelry, make-up, and accessories, if appropriate for them) to their laboratory session. Instructions were provided via email at the time of recruitment, then emphasized verbally on a scripted telephone call conducted one day prior to the experiment. The instruction email read: "Bring one entire outfit of your choice to your laboratory session that makes you feel attractive [comfortable]. This clothing should be consistent with what you would wear if you were going

out to a party or spending or night out with someone you were romantically interested in [spending a relaxed day at home or with friends]". On the telephone call we emphasized that participants in the beautification condition should bring an outfit and accessories that they would wear on a "hot date", and control participants should bring an outfit and accessories (if appropriate for them) that they would wear relaxing at home with friends. We left voicemails and sent text messages to participants we were unable to reach verbally.

After arriving at the laboratory, participants were told that they would spend 10–15 minutes inside a private change-room to allow them to 'get into character' as if they were dressing for a hot date [a day at home with friends]. Inside the change-room, participants were provided a range of beauty tools including sterilized make-up (eye-shadow, lip-gloss, lipstick, blush), make up removal wipes, hair accessories (bobby pins, headbands, hair ties, hair clips, hair-spray, dry shampoo), and a full-length mirror. We told all participants to use these beauty tools as well as their own beauty supplies only if it helped them get into character according to their condition instructions. We emphasized in both conditions that if they would not usually use beauty supplies to adjust their appearance for a hot date [a day at home with friends] then they should not use them now. Likewise, if they would usually remove their make-up for a day at home with friends [a hot date] then they should remove their 'makeup now (and they could reapply it at the end of the session).

All instructions were identical in both conditions so participants were free to act according to their normal behavior in preparation for a hot date [a day at home with friends]. We provided all participants 10–15 minutes for the task and told them to leave the change-room whenever they felt they had fulfilled the condition instructions. Participants who finished early waited until 10 minutes had elapsed then began the next part of the study (and were informed of this protocol prior to entering the change room). Participants also completed a filler task and a 4-item assertiveness questionnaire 30 after the beautification manipulation. The filler task allowed us to standardize the timing of the estradiol measurement.

**Manipulation checks.** Upon completion of the experiment, participants answered the extent to which their outfit made them feel relaxed [attractive] and represented something they would wear while sitting at home on the couch [to a party or out on a date] (1 = not at all, 7 = very much). We also took photographs of participants in their outfits and three coders blind to condition rated the photographs on sexual attractiveness striving ("How sexually attractive is this person trying to be to others?"; 1 = not at all, 7 = very; $ICC$ (2,3) = .90, $F$(143, 286) = 9.99, $p < .001$).

**Explicit assertiveness.** Thirty-four people in a pilot study read a description of the concept of assertiveness then scored 40 traits on assertiveness then desirability on two 7-point scales (1 = very non-assertive/undesirable, 7 = very assertive/desirable; from 30). One-sample $t$-tests confirmed that 15 traits were significantly higher than the scale mid-point ($ps \leq .001$). The traits were: clever, creative, conscious, single-minded, pushy, controlling, autonomous, self-centered, intentional, relentless, efficient, competent, dominant, active, and independent. To measure explicit assertiveness, participants rated the extent to which each of the 40 traits represented them on the day of testing. We then computed partial within-subject correlations between the self and these traits by correlating participants' ratings for the 15 traits with the assertiveness scores of those traits from the pilot study, controlling for their desirability scores ($M_r$ = -0.17, $SD_r$ = 0.34, range $r$ = -0.72 to 0.75). We controlled for the traits' desirability to minimize the effect of positive affect on this measure.

**Sexual motivation.** Participants rated the extent to which the traits 'seductive', 'flirtatious', 'sexually open', 'sensuous', and 'promiscuous' characterized them on the day of testing (1 = not at all, 7 = very much). Reliability was acceptable and scores were averaged ($\alpha$ = .80, $M$ = 3.02, $SD$ = 1.26).

**Positive affect.**    In the Expanded Positive and Negative Affect Schedule [31], participants indicated the extent to which they felt 60 words and phrases that describe different emotions (e.g., bold; 1 = very slightly or not at all, 5 = extremely). We computed the general positive affect subscale for our measure of positive affect [31].

**Implicit assertiveness.**    We used a Single Category Implicit Association Task to measure implicit assertiveness [30]. Twenty-six people read a description of the concept of assertiveness then rated 61 traits on assertiveness on a 7-point scale (1 = very non-assertive, 7 = very assertive). High assertiveness was characterized as an ability to assert and expand one's self, affect one's environment, and perform actions. Low assertiveness was characterized as unassertiveness. The top and bottom eight traits formed the high and low assertiveness words for the SC-IAT and showed significant group differences on assertiveness, $t(25) = 7.95$, $p < 0.001$, $M_{\text{LowAssertiveness}} = 2.29$, $SD = 1.18$, $M_{\text{HighAssertiveness}} = 5.59$, $SD = 1.52$, $d = 1.48$. The high assertiveness words were: decisive, driven, go-getter, self-aware, persistent, independent, productive, and strong-minded. The low assertiveness words were: dependent, meek, hesitant, apathetic, idle, inactive, unconcerned, and scatterbrained.

Participants categorized these traits with seven words characteristic of the self (me, my, mine, self, myself; and the participant's own first name and nickname [or, if they had no nickname, their name again]). Implicit associations were assessed by asking people to press the same response key for low assertive + self and to press the opposite response key for high assertive + self. These associations were then reversed and the order in which participants performed these trials was counterbalanced. The SC-IAT effect is the difference in response latency between low assertive + self and high assertive + self. We calculated $D$ scores following 32 [32] and higher scores indicate stronger associations of the self with assertiveness.

**Assertive behavior.**    Of the entire sample, 20 participants were not able to participate in the assertive behaviour task because they arrived late to their experimental session and it ran over time (and another participant was waiting to start their session). Of the remaining 125 participants, 63 agreed to participate. They were given a blank worksheet to prepare a 30 second monologue response to the generic question: "Why are you a competent employee and a good choice for a job where you will lead others?" We chose this question to measure assertive behavior because competence is a key dimension of agency [14,15], and the task required that participants assert their competence and worth in the job interview. We told participants that the video would be shown to others, but in reality, all videos were converted to audio-only for coding judgements. We did not specify the sex of the person judging the job interview tapes, and all experimenters were female. Coders rated the audio clips without video so we could remove the potential confound of clothing on coders' judgements of women's assertiveness [33], but retain the effect of the manipulation on assertive behavior during the interview.

Participants had a maximum of five minutes to prepare their monologue and sat facing the video camera to present it. All participants were allowed to record multiple takes if desired, but the vast majority of participants completed one take only. The monologue was coded by one woman and one man, both blind to condition. Raters used a 7-point scale (1 = not at all, 7 = very much) to rate assertiveness ("How assertive is this person?" then "To what extent does this person strive to affect their environment?"). Agreement between coders was strong and items were averaged, $ICC (2,2) = .94$, $F(62, 186) = 16.12$, $p < .001$, $M = 4.75$, $SD = 1.29$.

To confirm that the women who elected to complete the assertive behavior task were not a distinct subgroup, we conducted a series of one-way ANOVAs comparing those who opted to take part in the job interview with those who did not (controlling for condition) on all the variables in Table 1. There were no significant differences for any variable between women who opted to take part in the job interview and those who did not ($p$s $\geq .138$). We also examined whether beautification affected women's willingness to participate in the job interview (i.e., a

**Table 1. Differences between control and beautification conditions on the dependent variables in Experiment 1.**

| Variable | Mean (SD) | | *T* | *df* | *p* | *d* |
|---|---|---|---|---|---|---|
| | **Beautification** | **Control** | | | | |
| Explicit assertiveness | -0.09 (0.35) | -0.24 (0.32) | 2.64 | 143 | .009** | 0.44 |
| Implicit assertiveness | 0.03 (0.18) | -0.04 (0.14) | 2.48 | 140 | .015* | 0.41 |
| Assertive behavior | 4.90 (1.27) | 4.64 (1.31) | 0.81 | 61 | .421 | 0.21 |
| Sexual motivation | 3.36 (1.25) | 2.73 (1.20) | 3.10 | 143 | .002** | 0.60 |
| Positive mood | 3.03 (0.69) | 2.75 (0.69) | 2.49 | 143 | .014* | 0.41 |

Note.

† $p < .10$.

* $p < .05$.

** $p < .01$.

*** $p < .001$.

selection effect). The effect of the manipulation was not significant, $\chi^2(1, N = 125) = 0.01$, $p = .94$, meaning that women were equally likely to participate in the job interview in both conditions.

## Data analysis

To investigate the effect of beautification on women's assertiveness, we first ran a series of independent samples *t*-tests on the dependent variables. To test whether the effects of beautification were robust to the inclusion of positive affect, we regressed explicit and implicit assertiveness onto condition and positive affect. We excluded standardized residual outliers greater than or equal to ± 2 for regression analyses (<4.22% of cases), and *z*-score standardized all predictors; none were significantly skewed ($ps \geq .109$). Zero-order correlations are in Table A in S1 Appendix.

## Results

### Manipulation checks

Ninety-five percent (94.7%) of control condition participants and 98.4% of beautification condition participants indicated that the outfit they wore was consistent with their condition instructions. Eight participants who scored below the mid-point were excluded from further analysis (as noted in the Participants section). An independent samples *t*-test confirmed that outfits in the beautification condition ($M = 5.17$, $SD = 0.96$) gave the impression of significantly greater sexual attractiveness striving than those in the control condition ($M = 3.40$, $SD = 1.04$) and that the corresponding effect size was large, $t(142) = 1.51$, $p < .001$, Cohen's $d = 1.77$.

### Beautification effects

There were significant differences between conditions on implicit and explicit assertiveness, as well as sexual motivation and positive affect; see Table 1. Compared to women in the control condition, women in the beautification condition explicitly reported that high assertiveness words described them better and implicitly associated themselves more strongly with assertiveness. Participants in the beautification condition also reported more sexual motivation and positive affect than women in the control group. There was no direct effect of beautification on assertive behavior in the mock job interview.

Three control participants chose pajamas as the clothes that they would wear at home with friends. As a sensitivity analysis, we re-ran all analyses excluding these women; Effect sizes and significance levels were consistent with those reported herein. For the 4-item assertiveness questionnaire, a grammatical mistake meant that this questionnaire asked participants about their past assertive behavior (on days prior to the experiment) instead of their assertiveness on the current day. Beautification did not affect reports of past assertive behavior ($p = .478$, $d = 0.11$).

### Beautification effect after including positive affect (sensitivity analysis)

The regression of implicit assertiveness onto condition and positive affect showed that the effect of beautification on implicit assertiveness ($\beta = .05$, $p = .032$) was robust to the inclusion of positive affect ($\beta = .003$, $p = .828$). Likewise, the regression of explicit assertiveness onto condition and positive affect showed that the effect of beautification on explicit assertiveness ($\beta = .15$, $p = .004$) was also robust to the inclusion of positive affect ($\beta = .08$, $p = .039$), though positive affect did significantly predict higher explicit assertiveness. Neither condition ($\beta = .14$, $p = .676$) nor positive affect ($\beta = .22$, $p = .331$) predicted behavioral assertiveness.

### Exploratory mediation analyses for assertive behavior

We used the PROCESS macro model 4 [34] to run nonparametric bootstrapping analyses with 10,000 resamples to test mediation effects. In these analyses, mediation was significant if the 95% bias-corrected and accelerated confidence intervals (CI) for the indirect effect did not include zero [34]. We found that the effect of beautification on assertive behavior was mediated by sexual motivation (Fig 1). To the extent that the beautification manipulation increased sexual motivation ($\beta = .23$, $p = .045$), women demonstrated more assertiveness in their job interviews, indirect effect = .07, $CI_{95}$ [.002, .23]. The direct effect of sexual motivation on assertive behavior was significant and positive ($\beta = .32$, $p = .027$), and the direct effect of beautification ($\beta = .03$, $p = .815$) was not significant. As a sensitivity analysis, we added positive affect to the regression model as a covariate of assertive behavior. The indirect effect of beautification on assertive behavior via sexual motivation was still significant, indirect effect = .07, $CI_{95}$ [.002, .23], and positive affect was not a significant predictor of assertive behavior ($\beta = .08$, $p = .549$).

## Discussion

After beautifying their appearance, women demonstrated higher explicit and implicit assertiveness. Beautification also affected assertive behavior, but this effect was contingent upon the degree to which beautification increased female sexual motivation. All effects were independent of positive affect, suggesting that assertiveness induced by beautification is not reducible to elevated positive mood.

One limitation of Experiment 1 was that our investigation of the relationship between sexual motivation and assertiveness was exploratory in nature. This exploratory intent raised questions about replicability. We thus conducted a conceptual replication to test for effects of beautification on assertiveness. In this study, we examined beautification effects using a within-subjects design.

## Experiment 2

In Experiment 2 we measured assertiveness in women at two time points in their own homes: once when they wore whatever they wanted, then again after they extensively beautified their appearance (again, in the context of romantic attraction). We used the same explicit

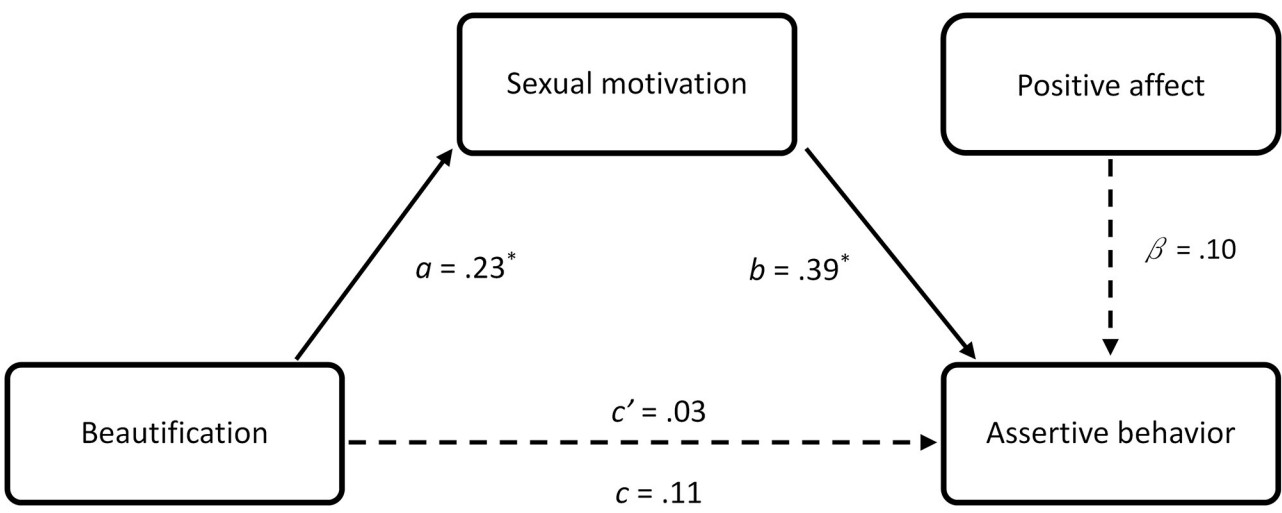

**Fig 1. Indirect effect of beautification on assertive behavior in Experiment 1.** Note. †p < .10. * p < .05. ** p < .01. *** p < .001. All coefficients are *z*-score standardized. Dashed paths are not significant, *n* = 63.

assertiveness measure as in Experiment 1, but added an additional task measuring assertive behavioral intentions in three consumer domains: public consumer assertiveness in an appearance-related domain; private consumer assertiveness in an appearance-related domain; and consumer assertiveness in an appearance-unrelated domain. We hoped to determine whether beautification increased assertiveness in appearance-related as well as appearance-unrelated domains, and included public and private contexts because consumer behavior shifts when people believe they are under public scrutiny [35]. We also measured sexual motivation after beautification, to reproduce and extend the positive association between sexual motivation, beautification, and female assertiveness observed in Experiment 1.

## Method

### Procedure

Women registered into a two-session online study investigating the effect of clothing on psychology. The order of the sessions was fully counterbalanced and there was a minimum three-day lapse between sessions. At the baseline session, participants completed measures of assertiveness and assertive behavioral intentions in whatever clothing they were wearing at the time, then completed a measure of trait self-objectification. At the experimental session, participants spent 15–20 minutes extensively changing their outfit (their entire clothing outfit, makeup, jewelry, hairstyle, shoes) into one that made them feel attractive. They then completed the same assertiveness and assertive behavioral intentions measures as the baseline session, as well as a state measure of sexual motivation.

To ensure participants followed the experimental instructions, they submitted a self-portrait "selfie" photograph of whatever they were wearing before being shown the beautification manipulation instructions. They were then shown the beautification manipulation instructions and submitted a second selfie after fulfilling the instructions and enhancing their appearance (15 to 20 minutes later). In both photographs, participants were instructed to hold up a randomly generated code-word in the photograph to ensure the photographs were taken during the experiment. Code-words automatically regenerated after 20 minutes. All photographs were reviewed by our research team for compliance (see "Beautification compliance check").

All data, materials, and correlations between variables are available on the OSF (https://bit.ly/2J0myLP) and all measures and manipulations are disclosed below.

## Participants

One hundred and thirty women ($M_{age}$ = 21.75, $SD$ = 4.46) were recruited from the University of New South Wales Sydney first year psychology student pool (who participated for course credit) or a paid psychology pool of students from University of New South Wales Sydney and the London School of Economics (who participated for AUD$15). Data were excluded from two participants who did not leave sufficient time between completing both sessions, eight whose photos did not allow us to assess compliance; and 26 women who failed the beautification manipulation check (see below). We also excluded two women who were homosexual because lesbian beauty norms differ from heterosexual beauty norms [36], and we aimed to reduce this source of variance in the study. At the time of data collection, a limitation of our university's online participation system was that participating in both parts of a two-part study was completely optional; participants could register and attend Part 1 even if they had no intention of completing Part 2. Thus, an additional 52 participants were withdrawn because they only registered for one session, leaving $n$ = 40 women ($M_{age}$ = 21.55, $SD$ = 4.92). A power analysis in GPower 3.1.9.2 indicated that the remaining sample size provided statistical power at 0.93 to detect a medium effect in a two-session repeated design ($f$ = .25; $a$ = 0.05) with two orders (baseline first, experimental first). Exactly 20% of the sample stated that their cultural background was East Asian, 17.5% were South-east Asian, 17.5% were Australian-New Zealand/Asian, 15.0% were Australian-New Zealand, 12.5% were Australian-New Zealand/European, 7.5% were European, and the remainder were North American, Central American, Southern Asian, Western Asian, or of mixed Asian cultures (7.5%).

## Materials

**Beautification manipulation instructions.** At the experimental session, participants were informed to: "Change your entire outfit into clothing that makes you feel attractive. By attractive, we mean the kind of thing you would wear if you were if you were going out on a hot date, spending a night out with someone you were romantically interested in, or going to a party where you might meet a potential romantic date. You must change your clothing, change your shoes, change your jewelry and accessories, as well as extensively change or "do" your make up (to suit you). Use the time to really "get into character", ensuring that you change your appearance just like how you would usually do if you were going on a date/out to a party. This is the most important thing about this experiment. We really want you to take the time so you feel your version of "very attractive"."

**Beautification compliance check.** To check compliance of the beautification manipulation, one research team member viewed all before and after selfie photographs from the experimental session, answering the following: "Is it conceivable that this participant spent 15 minutes altering their appearance to fulfil the condition instructions?" on a binary yes/no scale. As noted above, all participants for whom it was inconceivable that they had spent 15 minutes enhancing their appearance ($n$ = 26) were excluded from analysis. Retaining these women did not largely change the interpretation of results presented here (analyses are available on the OSF).

**Explicit assertiveness, sexual motivation, and self-objectification.** For explicit assertiveness and sexual motivation, we used the same measures from Experiment 1. We measured trait self-objectification using eight items from the Self-Objectification Questionnaire [37], each measured on a 7-point scale (e.g., "I rarely think about how I look", 1 = not at all, 7 = very

much; $a = .72$, $M = 3.24$, $SD = 0.97$). The score was $z$-standardized so that higher scores indicated more self-objectification.

**Assertive behavioral intentions.**   Three vignettes were created to measure assertive behavioral intentions in three consumer domains. The first vignette measured public assertiveness, operationalized via participants' willingness to revisit a shoe store to return a faulty item. The second vignette measured assertiveness that was private (i.e., the willingness to call a beauty salon whom they believed over-charged them to clarify the bill). The third vignette measured assertiveness in a non-appearance related domain, operationalized via their willingness to contest a bill from a mechanic who completed work on their automobile without informing them (neither public nor private was specified). Measures were on a 7-point scale (1 = not at all likely, 7 = very likely), with higher scores indicating more assertive intentions.

### Data analysis

A series of repeated-measures ANCOVAs tested whether beautification increased assertiveness, and whether the effect of beautification on assertiveness was moderated by trait self-objectification or state sexual motivation. We entered the main effect of time (Time 1: baseline session; Time 2: experimental session) into the ANOVA, along with z-standardized covariates of self-objectification and sexual motivation. Where interactions between time and self-objectification or sexual motivation were significant, we computed a difference score by subtracting baseline assertiveness scores from experimental assertiveness scores, such that high scores indicated more assertiveness after beautification. We then analyzed partial correlations between this difference score and self-objectification or sexual motivation, controlling for the other predictor. Zero-order correlations are in Table B in S1 Appendix.

### Results

Descriptive statistics and model results are in Table 2. For explicit assertiveness, the main effect of time was not significant and was not qualified by a time (Time 1: baseline session; Time 2: experimental session) × self-objectification interaction. However, there was a time × sexual motivation interaction, $F(1, 37) = 4.31$, $p = .045$, $\eta_p^2 = .10$. The partial correlation between the explicit assertiveness difference score and sexual motivation was $r_p (37) = .32$, $p = .045$, indicating that higher sexual motivation after beautification was positively correlated with increased explicit assertiveness.

For public consumer assertiveness, a main effect of time trended toward conventional levels of significance, $F(1, 37) = 3.63$, $p = .064$, $\eta_p^2 = .09$. Women reported a willingness to engage in somewhat greater public consumer assertiveness after beautification, see Table 2, and there was no time × sexual motivation or time × self-objectification. For private assertive consumer behavior, the main effect for time was not significant, but there was a time × self-objectification, $F(1, 37) = 5.70$, $p = .022$, $\eta_p^2 = .13$), and a marginally significant time × sexual motivation interaction $F(1, 37) = 3.66$, $p = .063$, $\eta_p^2 = .09$). Partial correlations indicated that women were more likely to report a willingness to engage in private consumer assertiveness after beautification if they were higher on trait self-objectification, $r_p (37) = .37$, $p = .022$, and to some extent, if they felt more sexually motivated, $r_p (37) = .30$, $p = .063$. For consumer assertiveness in an appearance-unrelated domain, neither the main effect of time nor the time × self-objectification and time × sexual motivation interactions were significant.

### Discussion

Using a within-subjects design, we found that beautification increased explicit assertiveness to the extent that beautification increased women's sexual motivation. For assertive consumer

**Table 2. The effect of beautification (time), sexual motivation, and self-objectification on explicit assertiveness and assertive behavioral intentions in three domains in Experiment 2.**

| Variable | Predictor | Baseline M (SD) | Exp M (SD) | F | p | $\eta_p^2$ |
|---|---|---|---|---|---|---|
| Explicit assertiveness | Time | .07 (.36) | .08 (.38) | 0.04 | .844 | < .01 |
| | Time × Sexual motivation | | | 4.31 | .045* | .10 |
| | Time × Self-objectification | | | 0.11 | .739 | < .01 |
| Public consumer assertiveness | Time | 3.48 (1.74) | 3.90 (1.77) | 3.63 | .064† | .09 |
| | Time × Sexual motivation | | | 1.09 | .302 | .03 |
| | Time × Self-objectification | | | 0.96 | .335 | .03 |
| Private consumer assertiveness | Time | 4.38 (1.85) | 4.70 (1.71) | 1.95 | .171 | .05 |
| | Time × Sexual motivation | | | 3.66 | .063† | .09 |
| | Time × Self-objectification | | | 5.70 | .022* | .13 |
| Appearance unrelated consumer assertiveness | Time | 5.35 (1.55) | 5.50 (1.47) | 0.54 | .467 | .01 |
| | Time × Sexual motivation | | | 0.83 | .369 | .02 |
| | Time × Self-objectification | | | 0.02 | .869 | < .01 |

Note.

† $p < .10$.

* $p < .05$.

** $p < .01$.

*** $p < .001$.

Exp = experimental. All degrees of freedom are (1, 37). The means are estimated marginal means, controlling for covariates evaluated at the mean.

behavioral intentions, findings were mixed. Beautification had a direct effect on increasing willingness to endorse public consumer assertiveness, but the effect did not reach conventional levels of statistical significance. Beautification also elevated endorsement of private consumer assertiveness, but the effect was moderated by trait self-objectification. The more women tended to self-objectify, the more they reported willingness to engage in private consumer assertiveness after beautification. This effect was also moderated by sexual motivation, showing the same pattern. We found no effect of beautification, self-objectification, or sexual motivation on consumer assertiveness unrelated to appearance.

## General discussion

Research derived from objectification theory has emphasized the negative consequences of beautification and related practices, highlighting that they harm women and are derived from a cultural context that disempowers them [1,5–7,24]. An alternative perspective, derived from sociometer theory, holds that beautification can benefit women by raising their self-esteem in important domains [8,11,12]. We added clarity to this research area by experimentally manipulating beautification through within- and between-subject designs, and subsequently measuring multiple indicators of assertiveness, as well as positive mood, sexual motivation, and self-objectification. Our results suggest that beautification can increase assertiveness in women, but that the effect may be domain-specific. These findings shed light on a key tension in female psychology by challenging the notion that beautification and related appearance-enhancing phenomena are necessarily disempowering.

Many of our effects were dependent on beautification increasing sexual motivation, with beautification elevating assertiveness only when it also elevated sexual motivation. This finding is consistent with previous research [30], and suggests that the effect of beautification on assertiveness depends upon the degree to which beautification increases the subjective feeling of

sexual attractiveness. By including measures of behavioral assertiveness (Experiment 1) and assertive behavioral intentions in three domains (Experiment 2), we intended to distinguish whether beautification-induced assertiveness was domain-specific or domain-general. Unfortunately, results were inconclusive: We did not find a significant effect for beautification in our appearance-unrelated consumer assertiveness vignette, however, we did find that beautification increased assertive behavior in the mock job interview in Experiment 1 (to the extent that it also increased sexual motivation). Future work teasing out these effects would help to clarify the conditions under which beautification can increase assertiveness, and whether that increase is specific to the appearance domain, or whether effects might transfer to unrelated domains.

Beautification interacted with sexual motivation to increase explicit assertiveness in women, regardless of trait self-objectification. Surprisingly, trait self-objectification was positively associated with a beautification-induced willingness to act assertively in one of our vignettes. This finding is supportive of parallel work showing that self-objectification and its antecedents can raise women's self-esteem in particular contexts [11,12]. Though these effects warrant replication, they suggest that conceiving of self-objectification as an entirely deleterious phenomenon may mischaracterize its psychological effects. The degree to which self-objectification may translate into enhanced female empowerment in some conditions is perplexing, yet it is also a worthwhile topic for future research.

## Implications for understanding self-objectification

These results provide further insight into understanding women's motivation for appearance-modifying behaviors, including self-objectification and self-sexualization. Many of these phenomena are motivated by desires to elevate attractiveness to new or existing romantic partners [38,39]. However, our findings suggest that women may also engage in these behaviors to increase assertiveness as well as mood. Thus, a desire for feeling empowered may partially account for women's beautification practices and consumption of appearance-enhancing products. This conception offers a unique perspective on why women are more beauty-focused when the economy declines (the lipstick effect; [40]). Beautification may provide an affordable way to elevate the subjective experience of empowerment in ecological conditions that often constrain agentic action [41].

The negative effects of self-objectification—including usurping women's attentional and cognitive resources and increasing the likelihood of mental health problems—usually result from intermediary processes, such as elevated body shame and body surveillance [3]. Our findings raise the possibility that beautification may not always elicit these intermediary processes, and our work suggests that beautification can elicit sexual motivation as well. A defining difference between whether beautification and related phenomena empower or disempower women, then, may depend upon which intermediary processes are elicited. For example, if beautification elicits appearance anxiety or body shame, it may reduce assertiveness; If beautification elicits sexual motivation or high self-esteem, it may heighten assertiveness. Future investigation into the intermediary processes induced by appearance-relevant behaviors on positive and negative psychological outcomes would be a welcome contribution to future work.

Contextual effects—such as the person a beautified woman believes is judging her [42]—are also likely to be important. We focused on beautification in one situation only, and it is unclear whether mandatory beautification in other contexts (e.g., stipulated by an employer for an important meeting) would show similar effects. Women often become targets of backlash when they act assertively, especially in domains that are stereotype-inconsistent [43], and attractive women may be especially likely to be targeted. Women who engage in beautification

and appearance-enhancing phenomena can also become targets of aggression by others, men and women like [33,44–47]. Thus, although increases in beautification may engender benefits to women, in certain contexts it may also engender costs. The contexts under which women may express assertiveness and beautify without suffering backlash effects, or the contexts under which women experience beautification as especially disempowering are important future research topics.

## The paradox of sexualized beautification and female agency

Although the current work provides evidence for a conditional effects of beautification on female assertiveness, our findings appear to be inconsistent with work showing that men and women perceive that women in attractive, revealing clothing lack agency [33,48,49]. Why is it that people perceive that women in such clothing lack agency, whereas the women themselves may potentially feel and behave in a more assertive manner? Compelling evidence demonstrates that people derogate those who act counter to the status quo [50]. Perceptions that women who engage in beautification lack agency may thus function to penalize women who threaten notions of demure and passive femininity through asserting sexual power [43,51]. Perceiving that these women lack agency may also support male dominance by discrediting the agency that some women demonstrate via beautification.

Equating beautification or self- sexualization with low agency may also reflect the cultural suppression of female sexuality, an ever-present albeit culturally variable phenomenon that sanctions women's sexual self-expression more heavily than men's. Although the drivers of the cultural suppression of female sexuality remain controversial [52–54], evidence supports the idea that competition between women can encourage them to suppress the sexuality and attractiveness-enhancing efforts of other women. Derogating such women as cultural dupes, who misunderstand female agency and how they are perceived by others, may thus function to reduce the occurrence of competition amongst women by elevating anxiety in potential competitors. Ultimately, such a process may function to diminish the threat of another woman's physical and sexual attractiveness.

Perceptions that sexualized women lack agency may also function to motivate sexual approach in men. Evidence suggests that some men find cues of sexual vulnerability and low agency in women to be alluring [55]. From a functional perspective, perceiving low agency in such women may be attractive to men because it reduces the threat of rejection, female infidelity, and paternity uncertainty associated with female sexual agency. It is also plausible that low agency women are perceived as less likely to rebuff sexual advances and easier to monopolize [30]. For these reasons, men's perceptions of low agency in women may be a cognitive bias that engenders sexual approach, akin to the robust bias men show to over-estimating women's sexual intent [56,57]. Future work investigating these notions would provide valuable insight into the constancy of patriarchal culture over time and provide. Research could also clarify the paradoxical nature of men's views of women's agency, and women's view of their own agency.

## Limitations and future directions

We aimed to provide a rigorous test of the effects of beautification on assertiveness by employing explicit, implicit, and behavioral indicators of assertiveness, ecologically valid designs, and by testing the importance of theoretically relevant mechanisms and confounds (i.e., sexual motivation, positive affect). That being said, our findings are limited in several ways. Although patterns of variation in Experiment 2 were generally consistent with Experiment 1, two effects from Experiment 2 did not reach conventional levels of statistical significance. Likewise, in Experiment 2, we failed to replicate the direct effect of beautification on explicit assertiveness,

finding instead that the effect was dependent on beautification eliciting sexual motivation. This latter finding highlights the importance of sexual motivation to the beautification–assertiveness link, but it weakens our ability to draw conclusions about the overall relationship between the two phenomena. Likewise, whether assertiveness effects are domain-general, or specific to appearance-relevant domains only, was unresolved by the current work. Based on sociometer theory, we speculate that beautification-induced assertiveness may be strongest in appearance-related domains, and weaker, albeit present in other domains.

A further limitation is that the instructions in the beautification condition were multi-faceted. The instructions informed women to dress for a night out where they might meet someone they were romantically interested in, a hot date, and a party. We emphasized "hot date" in the verbal instructions most frequently both before and during the experimental sessions, and parties are locations where young people commonly meet romantic partners. We did so because the aim of our study was to focus on beautification in the context of romantic relationships, and attractiveness in the domain of romantic relationships is a domain where women are especially likely to derive self-esteem [8,11]. The multi-faceted nature of these instructions; however, may have introduced unnecessary noise in our experimental manipulation, weakening our ability to detect effects.

Another limitation is that design differences between Experiments 1 and 2 may account for some variability in our findings. Experiment 1 occurred in the laboratory, meaning that participants were seen by the experimenter after they changed their clothing and makeup. In contrast, Experiment 2 occurred online, and participants completed the experimental session in their home. We utilized this design difference so participants in Experiment 2 had the choice of their entire wardrobe and all of their own beauty products at their disposal. Unfortunately, this distinction between public and private may have weakened findings in Experiment 2. It is possible that the element of being seen in public after one enhances their sexual appearance strengthens the effect of beautification on female assertiveness, resulting in stronger effects in public versus private settings. Such an interpretation would account for weaker effects in Experiment 2 compared to Experiment 1.

A final limitation is that we only controlled for one individual difference in our analyses. Although trait self-objectification was highly relevant, many other individual differences affect women's willingness to beautify, self-objectify, and self-sexualize. For example, recent work indicates that ideological components related to higher order personal values are especially relevant [58]. Testing whether findings reported here are sensitive to these differences, and the individual differences predictive of beautification, would strengthen our conclusions.

## Conclusion

We found positive effects of beautification on women's implicit, explicit, and behavioral assertiveness. Some of these effects depended on beautification increasing sexual motivation. The current work challenges the assumption that beautification is necessarily disempowering to women. It provides novel insights into the positive psychological effects of beautification on women, highlighting the conditional relationships between beautification, sexual motivation, and expressions of female assertiveness.

## Supporting information

**S1 Appendix. Appendix for Experiment 1 and Experiment 2.**
(DOCX)

## Author Contributions

**Conceptualization:** Khandis R. Blake, Robert Brooks, Thomas F. Denson.

**Data curation:** Khandis R. Blake, Lindsie C. Arthur.

**Formal analysis:** Khandis R. Blake, Thomas F. Denson.

**Investigation:** Khandis R. Blake.

**Project administration:** Khandis R. Blake.

**Resources:** Robert Brooks, Thomas F. Denson.

**Supervision:** Khandis R. Blake.

**Writing – original draft:** Khandis R. Blake, Robert Brooks.

**Writing – review & editing:** Khandis R. Blake, Robert Brooks, Lindsie C. Arthur, Thomas F. Denson.

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
