## [Decision Letter · Decision Letter 0]

1 Oct 2019

PONE-D-19-16746

Sexually Motivated Beautification Can Increase Assertiveness in Women

PLOS ONE

Dear Dr Blake,

Thank you for submitting your manuscript to PLOS ONE. After careful consideration, we feel that it has merit but does not fully meet PLOS ONE’s publication criteria as it currently stands. Therefore, we invite you to submit a revised version of the manuscript that addresses the points raised during the review process.

Whilst one reviewer suggested minor revisions, the second reviewer has more substantial concerns and thus recommends a major revision.

I enjoyed reading your manuscript, but as I share some of the concerns, I invite you to submit a revised version that addresses the points raised in the reviews.

Similar to Reviewer 2, one of whose main points concerns the theoretical foundation, I think that a more comprehensive framework will help you reach the main goal of your study stated in the introductory paragraph. In essence, you attempt to show that women’s preoccupation with their physical appearance may not inevitably present a paradox by investigating the conditions under which beautification may increase assertiveness in women. In other words, although it has been shown that it can be a harmful, potentially disempowering pressure on women, beautification may yield positive outcomes in certain circumstances.

Given the important contribution such findings might make to current scientific and societal debates, a considerably more careful theoretical treatment of the antecedents, processes, and outcomes of beautification is warranted.

For instance, an important question arises in light of Choi and DeLong’s (2019) tenet that “sex appeal” had become so greatly valued in Western society that young women voluntarily imposed sexualisation to the self and of Rollero and De Piccoli’s (2017) findings that personal values (e.g., self-enhancement, self-transcendence, openness to change) influence the degree to which people self-objectify. Notwithstanding a discussion of the relationships between self-objectification and self-sexualisation (see Reviewer 2’s comments on the need of putting your main concepts in perspective), this suggests that person variables beyond trait self-objectification may influence the beautification-assertiveness relationship, but that do not feature in your current theory section.

Pointing to potential context effects, Yilmaz and Bozo’s (2019) experimental study of women’s state self-objectification, body shame, and negative mood suggests that the processes underlying beautification effects seem to be rather complex. In that study, women’s own look at their body in a swimming costume impacted self-objectification and negative mood significantly more than being looked at by other women or men and there wasn’t a difference between female and male gazes; differences became insignificant, however, for the same women wearing jeans and jumper. Discussing the system of such context effects could help address Reviewer 2’s concerns that you chose an ecologically valid way to assess assertiveness, given that the participants were “compelled to beautify by the experimenters”. I concur that contextual constraints most likely play a substantial part although I wouldn’t see a problem with your experimental manipulation. What assertiveness effects would one expect, however, if beautification were ‘mandatory’, e.g. if stipulated by a women’s employer for an upcoming important client meeting?

In sum, a more detailed discussion of the interplay of person and context factors would help specify the conditions that co-determine the potential pressurising and empowering facets of one’s preoccupation with physical appearance and beautification.

I am confident that you will find both reviewers’ comments helpful in improving your manuscript.

We would appreciate receiving your revised manuscript by Nov 09 2019 11:59PM. To enhance the reproducibility of your results, we recommend that if applicable you deposit your laboratory protocols in protocols.io, where a protocol can be assigned its own identifier (DOI) such that it can be cited independently in the future. For instructions see: http://journals.plos.org/plosone/s/submission-guidelines#loc-laboratory-protocols

We look forward to receiving your revised manuscript.

Kind regards,

Christian Stamov Roßnagel

Academic Editor

PLOS ONE

Journal Requirements:

3. We note that several details from the Methods section have been blinded for peer review. Please note that PLOS ONE does not offer double blind peer review. We would be grateful if you could update your Methods section to include all the information previously redacted from the manuscript.

Please note that according to our submission guidelines (http://journals.plos.org/plosone/s/submission-guidelines), outmoded terms and potentially stigmatizing labels should be changed to more current, acceptable terminology. For example: “Caucasian” should be changed to “white” or “of [Western] European descent” (as appropriate).

4. Thank you for including your ethics statement:  "The UNSW Ethics Committee approved Study 1 (HREAP 2556) and Study 2 (HREAP 2906). Written consent was obtained for both studies.".   

a.Please amend your current ethics statement to include the full name of the ethics committee/institutional review board(s) that approved your specific study.

b.Once you have amended this/these statement(s) in the Methods section of the manuscript, please add the same text to the “Ethics Statement” field of the submission form (via “Edit Submission”).

5.  We note that Figure 1 includes an image of participants in the study.

Reviewers' comments:

Reviewer's Responses to Questions

**Comments to the Author**

1. Is the manuscript technically sound, and do the data support the conclusions?

Reviewer #1: Yes

Reviewer #2: Partly

2. Has the statistical analysis been performed appropriately and rigorously? 

Reviewer #1: Yes

Reviewer #2: I Don't Know

3. Have the authors made all data underlying the findings in their manuscript fully available?

Reviewer #1: Yes

Reviewer #2: Yes

4. Is the manuscript presented in an intelligible fashion and written in standard English?

Reviewer #1: Yes

Reviewer #2: Yes

5. Review Comments to the Author

Reviewer #1: PONE-D-19-16746

The present work, “Sexually Motivated Beautification Can Increase Assertiveness in Women” addresses the question of whether beautification can empower women to act assertively, doing so via two experiments. I find this research question to be an important one, and, to my knowledge, this work provides some of the first experimental evidence to address the authors’ focal question. Additionally, the work is generally well written, the experimental evidence is somewhat convincing, and the authors take care to not make greater claims than their data can or to sensationalize the claims they do make. I also appreciate the authors’ dedication to open science practices. This work is worthy of eventual publication in PLOS ONE. Below, I make some suggestions for improving the manuscript.

First, let me expand a bit on the important contributions that this work makes to the literature on women and female agency. I see this work as part of a nascent trend in research to question some of the assumptions that many people—especially those of us in liberal-leaning university settings—might hold about women’s behavior. For example, Andrea Meltzer’s recent work (2019, PSPB) challenges whether women are always harmed by being valued for their beauty (as many might assume they are), and Tarran, Muggleton, and Fincher (2019, EHB) challenge whether the suppression of women’s sexually is always enacted by men (as many might assume it is). Here, the authors question whether making oneself more physically attractive can be empowering for women (as many might assume it is not or should not be), and they use an experimental design to explore this. This is paper thus contributes to a larger trend that seeks to describe and understand how women actually think, feel, and behave (and not how many might assume women would or should think, feel, and behave).

This paper also engages with some feminist literature, and treats that work and related findings fairly. My main suggestion for improving the work is that the authors lean further into the two sides of the “feminist debate” that they mention on p. 3 in framing their Introduction. There is the possibility to really highlight the diverging perspectives—both with some empirical support—that beautification harms and/or empowers women. The two papers I mentioned above both highlight the two sides of their respective debates very nicely; they could be used as models, especially the former. Focusing the Introduction more the two sides of a debate—which their research question gets to the heart of—will make the paper a more enthralling read for both researchers and lay the public.

I would also ask that the authors take a bit more text to delineate agency, assertiveness, beautification and (self-)objectification. At first read, we all think that we know what each of these terms means, but they have some useful nuances. A merit of the paper is that is draws all of these terms together, and something as simple as a parenthetical definition—perhaps even from the work they are citing at the moment (e.g., Liss et al., 2011 on self-objectification)—would do.

I also share some more minor suggestions. Most of these suggestions are to add further nuance and/or further speculation as to the causes and consequences of the effect.

- It would be beneficial if the authors would speculate a bit more as to how and why beautification might have this effect. That is, what is the proximate and/or ultimate logic? For example, for women, beauty might be considered a source of power for reasons consistent with ultimate logic (a la Sell, Tooby, & Cosmides, 2009). This might also be a place to mention Leary’s (2005) sociometer work; that is, perhaps the effect lies in the fact that other people treat more attractive women more favorably—and/or women expect that other people will do so—which could influence women’s assertiveness.

- Related, I’m wondering whether it is the act of beautification, the feeling of having become more beautiful, and/or the expectation that others will find them more beautiful that is perhaps driving this effect.

- People might often assume that this effect is solely positive for women. (They might immediately think: A step toward gender equality maybe? Yay!). But increased beautification is likely to engender not only possible benefits (assertiveness), but also possible costs. Indeed, the authors discuss this in the GD (e.g., p. 24-25). There are additional costs they might note. For example, assertiveness might also have some costs, especially for women. Moreover, women who engage in beautification might also receive more aggression from other women (e.g., Leenaars, Dane, & Marini, 2008; Vaillancourt & Sharma, 2011).

- Related, for the discussion of beauty and agency on p. 26-27, see also Arnocky et al.’s (2019) recent Psychological Science paper.

- I would have expected the assertiveness to be a bit more domain-specific, mating-related, and primarily confined to less formal interactions with men. Some of the DVs from Experiment 2 (vignettes) and the use of a more formal interaction (i.e., the mock job interview) might begin to probe this a bit. (I’ll confess that I did not think the vignettes were the most solid operationalizations of private/public.) Then again, beauty might cue benefits beyond mating (e.g., Eisenbruch, Lukaszewski, & Roney, 2017), so perhaps it might also increase assertiveness beyond mating domains. I think some speculation regarding women’s audience(s) and to whom/in which situations this assertiveness might be most expected would be useful. (For the job interview, please provide a bit more information in the main manuscript. E.g., Was the interviewer male, an experimenter who simply gave the P the sheet, …?)

- As the author of a number of papers that focus on women’s psychology and behavior, I won’t require the authors here to speculate as to how this same process might play out among men; men were not their focus and for theoretically-sound reasons. (Still, in the West, men have increasingly become objects of the human gaze, and this might have serious practical implications for approximately 50% of the population.)

- One final minor comment (not minor suggestion): This paper, and the feminist debate with which it engages, calls to mind an interesting problem. Beautification might make women feel bad about themselves, might diminish women, might require women to spend finite time and money on hair products (meaning that time and money cannot be spent on “better” pursuits). But beautification might also make women more assertive in some instances. But then again, this increased assertiveness might owe to the fact that women live in a society where they are valued for their beauty, so is this “authentic” assertiveness? (The authors deal with this briefly on p. 5.) Who gets to say what is “authentic” assertiveness that “genuinely” empowers women? Do scholars? Do women themselves? Again, the experimental, empirical aspect of this paper provides meaningful contribution, in part by cutting through this.

Reviewer #2: Research reveals links between women’s beautification and a variety of outcomes, both positive and negative. The current work extends this empirical literature by testing for effects of women’s beautification on their assertive attitudes, behaviors, and intentions. Despite notable strengths of this work related to its aim and experimental rigor, I have concerns related to the theoretical foundation, how key constructs were operationalized, and potential factors that covary with the beautification condition. I detail these concerns and other questions, and provide suggestions that I hope will be helpful, below.

1. The authors frame this work as addressing a paradox related to women’s beautification efforts. Although this is a question of potential practical importance, the authors do not present a theoretical framework to guide the statistical tests presented within the manuscript. As is, it is not clear that the current tests were designed with resolving this paradox in mind. (For example, boundary conditions related to the potential positive vs. negative effects of these efforts are not considered). An organizational theoretical framework, or the presentation of competing theoretical frameworks, would bring clarity and structure to this work. An explicit theoretical framework might also address questions related to the selected measures and instruction sets, as described below.

In sum, the theoretical (and practical, see below) implications of this work should be more clearly elucidated. Without a clear theoretical framework, it is difficult to discern the importance of the present analyses. (For instance, analyses pertaining to individual differences in sexual motivation were presented as exploratory in Experiment 1, but become a focal analysis in Experiment 2. The theoretical and/or practical reason(s) for this change in perspective is not articulated.) This work would benefit from a more explicit specification of the guiding theoretical framework(s), and a clear delineation of which results would support (or refute) this view.

2. The conceptual overlap between measured constructs, as well as the interchangeable use of labels made it difficult to discern some of what the authors were attempting to assess and test. This broad concern was reflected in construct operationalization, measurement, and interpretation. For instance:

The authors use the terms “sexualization” and “beautification” nearly interchangeably throughout the manuscript. It is unclear if their primary hypotheses concern beautification effort or sexualization of one’s appearance (or perhaps sexual motivation). One might make different predictions regarding the potential effects of beautification versus sexualization on assertiveness (as well as the boundary conditions, moderating and mediating factors). This seems like a critical distinction to make, given this was the primary independent variable in the current work.

The authors also use the words agency and assertiveness in overlapping ways. It would be helpful to clearly operationalize the key outcome of interest in this work. Related to this point, it seems that the measure of positive affect used in Experiment 1 contains items that substantially overlap with feelings of assertiveness (e.g., confident, active, fearless, daring, strong, attentive, bold, lively, determined; NOT timid or afraid). It seems important to clearly define and operationalize key constructs, use consistent labels when referring to these constructs, and avoid overlap in the measurement of these constructs.

Finally, and related to Point #1, I wondered why the “behavioral intentions” assertiveness measure included in Experiment 2 focused exclusively on consumer domains. Assertive behavior, it is reasonable to assume, could be assessed across many domains. (That is, there seems to be no theoretical rationale for predicting that the enhanced assertiveness should only be expressed in consumer domains.) Further, it is unclear why the distinction between public vs. private, and appearance-related vs. not, was important to assess here.

3. I also had concerns and questions related to the instructional sets presented to participants (and subsequent participant exclusions) and potential confounds related to post-experiment plans.

Regarding the instruction sets, it is unclear why participants in the beautification/sexualization condition were prompted to ready themselves for a “hot date.” This seems likely to have primed mating motivations among most participants. This also seems methodologically-inappropriate if this work was intended to test the effects of beautification per se (rather than sexualization or mating motives) on assertiveness. Beautification efforts can certainly take place in the absence of explicit mating motivations. Related to this point, why were participants excluded based on non-heterosexual orientation? There is no explicit theoretical or empirical rationale provided for the relevance of women’s sexual orientation in understanding the link between beautification and assertiveness. (Non-heterosexual women also engage in appearance enhancement.) Further, I am unsure if this is an ecologically valid way to assess potential empowerment (or assertiveness) effects that might derive from women choosing to beautify, given that the female participants were compelled to beautify by the experimenters.

Across Experiments 1 and 2, I wondered if participants were told that they would be expected to return to their baseline state of attractiveness (e.g., “unbeautify”) before leaving (or ending the session). If participants were not required to do this, any effects that took place during the experimental session could be due to differences in plans among participants who engaged in beautification efforts vs. did not. It is reasonable to assume that participants who left the experiment wearing the outfits displayed in Figure 1, Panel A, would be more likely than the control participants to seek out social interactions with a variety of partners after the session (e.g., after investing this effort, they might have decided they would actually meet up with someone for a “hot date”). From this view, one could interpret any between groups differences as deriving from the thoughts, feelings, and expectations of women’s post-session plans. Of course, there is no way to know this was the case. However, this possibility cannot be ruled out if participants were not instructed to return to their baseline state.

4. Experiment 2 was presented as a replication of Experiment 1. However, there were important differences between these two experiments (beyond the within- vs. between-person design). First, as noted above, the analyses in Experiment 2 focused heavily on sexual motivation and self-objectification as moderators of the relationship between beautification and the varied measures of assertiveness. However, it is unclear why these moderators became a central focus. A clear theoretical rationale for these tests was not provided. Further, the analyses related to sexual motivation were presented as exploratory in Experiment 1 but were then emphasized in Experiment 2 based on the significant mediation analysis in Experiment 1, even though (again), a clear rationale for why this variable was important (other than the significant finding) was not presented. Finally, sexual motivation was tested as a mediator in Experiment 1 and a moderator in Experiment 2. If the point was to replicate Experiment 1, why wasn’t sexual motivation tested as mediating the relationships between beautification and assertiveness? (That is, sexual motivation should have been assessed at both timepoints, with the difference score tested as a mediator in Experiment 2.)

5. In addition to the dubious theoretical implications of this work, the practical implications could be debated. One might assume that assertive attitudes are practically important insofar as they motivate assertive behavior. Experiment 1 failed to find a direct effect of beautification on assertive behavior, and Experiment 2 implications were limited by an exclusive focus on consumer assertiveness.

6. In the abstract, the authors state that this work “provides important, novel insight into motivations that underlie women’s appearance-enhancing behaviors.” The current work does not provide much insight into the motivations underlying women’s attractiveness enhancement, given that women were explicitly instructed to beautify for a hot date. (That is, there is no variation in their motivation to self-beautify given that they all beautified in response to experimental instructions.)

7. Based on cell sizes, it appears that participant exclusions were not randomly distributed across conditions (i.e., there were fewer participants in the beautification condition in Experiment 1). Have the authors considered if (and how) this could have affected their results? For instance, is it possible that less assertive participants were less able or willing to follow the instructions in the beautification condition?

6. PLOS authors have the option to publish the peer review history of their article (what does this mean?). If published, this will include your full peer review and any attached files.

Reviewer #1: No

Reviewer #2: No

---

## [Author Response · Author response to Decision Letter 0]

25 Oct 2019

Please see Response to Reviewers document

---

## [Decision Letter · Decision Letter 1]

31 Jan 2020

In the context of romantic attraction, beautification can increase assertiveness in women

PONE-D-19-16746R1

Dear Dr. Blake,

We are pleased to inform you that your manuscript has been judged scientifically suitable for publication and will be formally accepted for publication once it complies with all outstanding technical requirements.

With kind regards,

Christian Stamov Roßnagel

Academic Editor

PLOS ONE

Additional Editor Comments (optional):

Reviewer #2 was unfortunately not available to review your revision. In light of your comprehensive and convincing revision and detailed responses to the reviewers' comments, we decided to accept based on Reviewer #1's recommendation and my own reading of your paper.

Reviewers' comments:

Reviewer's Responses to Questions

**Comments to the Author**

1. If the authors have adequately addressed your comments raised in a previous round of review and you feel that this manuscript is now acceptable for publication, you may indicate that here to bypass the “Comments to the Author” section, enter your conflict of interest statement in the “Confidential to Editor” section, and submit your "Accept" recommendation.

Reviewer #1: All comments have been addressed

2. Is the manuscript technically sound, and do the data support the conclusions?

Reviewer #1: Yes

3. Has the statistical analysis been performed appropriately and rigorously? 

Reviewer #1: Yes

4. Have the authors made all data underlying the findings in their manuscript fully available?

Reviewer #1: Yes

5. Is the manuscript presented in an intelligible fashion and written in standard English?

Reviewer #1: Yes

6. Review Comments to the Author

Reviewer #1: I won't go on needlessly here. I see this paper as fit for publication in PLOS ONE.

In my first review of this manuscript, I underscored the importance of this new brand of work that takes women as they are, in all their complexity, and not as anyone might wish they were. Women can be good, bad, ugly---just as can men. Women can aggress, seek status, and so on---just as can men. I continue to see this work as another step in the direction of better understanding female cognition and behavior---again, how woman are, regardless of how any theoretical, political, or ideological framework holds that they ought to be.

This paper does the important work that experiments must do: It uses the experimental method to test cause and effect. Moreover, related to my point above, it thus lets women's cognition and behavior speak for itself.

I appreciate the authors' responses to my comments. The authors did a thorough job in their revisions. For example, I believe the tweaked framing, which now sees them to pit against one another some ideas derived from objectification theory and some from a sociometer theory, has strengthened the paper.

I would hate for it to come across as if this short review were not extremely enthusiastic. Indeed, if other reviewers disagree, I'm happy to fight for the paper with much more text and evidence. Given the extensive attention that the authors paid to other reviewers' requests, I doubt that that will be the case.

7. PLOS authors have the option to publish the peer review history of their article (what does this mean?). If published, this will include your full peer review and any attached files.

Reviewer #1: No

---

## [Editor Report · Acceptance letter]

21 Feb 2020

PONE-D-19-16746R1 

In the Context of Romantic Attraction, Beautification can Increase Assertiveness in Women

Dear Dr. Blake:

I am pleased to inform you that your manuscript has been deemed suitable for publication in PLOS ONE. Congratulations! Your manuscript is now with our production department. 

With kind regards,

on behalf of

Dr. Christian Stamov Roßnagel 

Academic Editor

PLOS ONE